# Comparative Analysis of the Composition of Fatty Acids and Metabolites between Black Tibetan and Chaka Sheep on the Qinghai—Tibet Plateau

**DOI:** 10.3390/ani12202745

**Published:** 2022-10-13

**Authors:** Tongqing Guo, Xungang Wang, Qian Zhang, Lin Wei, Hongjin Liu, Na Zhao, Linyong Hu, Shixiao Xu

**Affiliations:** 1Northwest Institute of Plateau Biology, Chinese Academy of Sciences, Xining 810008, China; 2University of Chinese Academy of Sciences, Beijing 100049, China

**Keywords:** metabolic, meat, *longissimus dorsi*, grazing, pasture

## Abstract

**Simple Summary:**

Chaka sheep are adapted to a harsh, highly saline environment and are also known for having typical sensory characteristics attributed to free-range conditions and grazing on wild plants. Black Tibetan sheep are located in Guinan county, Tibetan Autonomous Prefecture of Hainan, Qinghai Province, and are well-adapted to the harsh plateau environment in China. In this study, we compared the fatty acids and metabolites in the *longissimus dorsi* muscle between Black Tibetan and Chaka sheep grazing in a highly saline environment. We found that upregulated organic acid and fatty acid biosynthesis increased the meat quality of Chaka sheep.

**Abstract:**

The objective of this study was to investigate and compare fatty acids and metabolites in the *longissimus dorsi* muscle between Black Tibetan and Chaka sheep grazing in a highly saline environment. A total of eight castrated sheep (14 months old) with similar body weights (25 ± 2.2 kg) were selected. The experimental treatments included Black Tibetan (BT) and Chaka sheep (CK) groups, and each group had four replications. The experiment lasted for 20 months. All sheep grazed in a highly saline environment for the whole experimental period and had free access to water. The results showed that the diameter (42.23 vs. 51.46 μm), perimeter (131.78 vs. 166.14 μm), and area of muscle fibers (1328.74 vs. 1998.64 μm^2^) were smaller in Chaka sheep than in Black Tibetan sheep. The ash content in the *longissimus dorsi* was lower in Chaka sheep than in Black Tibetan sheep (*p* = 0.010), and the contents of dry matter (DM), ether extract (EE), and crude protein (CP) in the *longissimus dorsi* showed no differences (*p* > 0.05). For fatty acids, the proportions of C10:0, C15:0, and tC18:1 in the *longissimus dorsi* were higher in Chaka sheep than in Black Tibetan sheep (*p* < 0.05). However, all other individual fatty acids were similar among treatments, including saturated fatty acids (SFAs), unsaturated fatty acids (UFAs), monounsaturated fatty acids (MUFAs), polyunsaturated fatty acids (PUFAs), and the ratios of n-6 PUFAs to n-3 PUFAs and PUFAs to SFAs (*p* > 0.05). A total of 65 biomarkers were identified between the two breeds of sheep. Among these metabolites, 40 metabolic biomarkers were upregulated in the CK group compared to the BT group, and 25 metabolites were downregulated. The main metabolites include 30 organic acids, 9 amino acids, 5 peptides, 4 amides, 3 adenosines, 2 amines, and other compounds. Based on KEGG analysis, eight pathways, namely, fatty acid biosynthesis, purine metabolism, the biosynthesis of unsaturated fatty acids, renin secretion, the regulation of lipolysis in adipocytes, neuroactive ligand–receptor interaction, the cGMP-PKG signaling pathway, and the cAMP signaling pathway, were identified as significantly different pathways. According to the results on fatty acids and metabolites, upregulated organic acid and fatty acid biosynthesis increased the meat quality of Chaka sheep.

## 1. Introduction

The Qinghai-Tibet Plateau is the largest and highest plateau in the world and is characterized by an extremely harsh environment, such as severe cold, hypoxia, and ultraviolet radiation [1]. Sheep (*Ovis arise*) that are indigenous to the plateau have adapted well to the prevailing conditions. However, the sheep cannot obtain enough food due to the long-term extensive grazing, breeding mode, lack of forage biomass, and low nitrogen in the pasture during much of the year [2]. In recent years, public awareness of safe and high-quality mutton has increased dramatically. Public health policies indicate that the intake of SFAs should be decreased and the consumption of PUFAs should be increased in humans [3]. Mutton, an excellent source of essential micronutrients, contributes to physiological and biochemical functions in humans [4]. Mutton from sheep on the Qinghai-Tibet Plateau is a highly consumed meat product due to its meat flavor, high nutritional value, low levels of contaminants, and high UFA and protein contents. These unique sensory qualities are closely related to the distinct environment and the natural grazing system [4].

The Chaka sheep, a geographical symbol of agricultural products used by the Chinese Ministry of Agriculture and Rural Affairs, is named after Chaka Salt Lake [5]. Chaka sheep are adapted to a harsh, highly saline environment and are also known to have typical sensory characteristics attributed to free-range conditions and grazing on wild plants. The unique natural environment has a cool climate, unpolluted air, unfertilized grazing pastures, and sufficient mineral water sources on the Qinghai-Tibet Plateau, which makes it ideal for sheep. Thus, we considered the possibility that mutton from Chaka sheep is positively associated with a harsh and highly saline environment. Black Tibetan sheep, also called Guide Black-Fur sheep, are located in Guinan county, Tibetan Autonomous Prefecture of Hainan, Qinghai Province, and are well-adapted to the harsh plateau environment in China [6]. They provide high-nutritional-quality meat and other necessities for the livelihood of local pastoralists on the Qinghai-Tibet Plateau. However, only a few studies have been conducted on these two different breeds, and studies have been conducted using different feeding systems.

The metabolic mechanism in muscle is complex, and the manipulation of muscle development, intramuscular fat deposition, and fatty acid composition for meat production is very important in sheep breeding [7]. Metabolomics is the study of metabolites, which are low-molecular-weight compounds in biological systems, and includes gas chromatography–mass spectrometry (GC-MS), nuclear magnetic resonance (NMR), and liquid chromatography–mass spectrometry (LC-MS). Metabolomics plays a crucial role in detecting nutrient, food, plasma, fecal, organic, and tissue metabolites in humans and animals [8,9,10,11]. Significant metabolites may also be used to explore novel biological pathways influencing the variation in meat flavor [8]. Studies of meat that used metabolomics have been associated with a combination of meat quality traits, focusing on specific factors associated with the animal’s genetic background and sensory scores [12]. However, very little is known about metabolites in both Black Tibetan and Chaka sheep in pastures.

Therefore, the objective of this study was to investigate and compare the fatty acids and metabolites in the *longissimus dorsi* muscle between Black Tibetan and Chaka sheep grazing in a highly saline environment and to obtain a basis for the meat quality of Chaka sheep.

## 2. Materials and Methods

### 2.1. Experimental Design

The research was conducted in Chaka Town, Wulan County, Qinghai Province, China (N36°45′34″, E99°20′35″, altitude of about 3125 m above sea level). It is classified as having a continental climate on the plateau. A total of eight castrated sheep (14 months old) with similar body weights (25 ± 2.2 kg) were selected, and the treatments included Black Tibetan sheep and Chaka sheep groups. There were four replications for each treatment. The experiment lasted 20 months, and sheep grazed in a highly saline environment for the whole experimental period. All sheep were provided with water ad libitum. Grazing began at 08:00 and ended at 18:00. The pasture species composition included *Elymus dahuricus*, *Allium chrysanthum*, *Poa annua*, *Polygonatum sibiricum*, *Astragalus laxmannii*, *Convolvulus arvensis*, *Achnatherum splendens*, and *Stipa capillata*. At the end of the experimental period, all sheep were slaughtered by exsanguination. Samples of the *longissimus dorsi* muscles were taken from the left side of the carcass and immediately vacuum-packed and frozen at −20 °C for the analysis of fatty acids and metabolites.

### 2.2. Analysis of Muscle Fiber Characteristics

The diameters, perimeters, and areas of myofibers in the *longissimus dorsi* were measured by traditional hematoxylin and eosin (H&E) staining. Slides were imaged using a Leica RM2235 slide scanner (Leica RM2235, Nussloch, Germany), and images were acquired using CaseViewer software (3DHistech, Ltd., Budapest, Hungary).

### 2.3. Analysis of Chemical Components in Mutton

The contents of DM, ash, and N were measured by the methods of the Association of Official Analytical Chemists (AOAC) [13]. The DM content was obtained by drying samples at 105 °C in a forced-air oven for 4 h. The content of ash was determined by complete combustion in a muffle furnace at 550 °C for 6 h. The N contents were assayed with a protein analyzer (K9840, Hanon Advanced Technology Group Co., Ltd., Jinan, China) with the Kjeldahl method, and CP was calculated as N × 6.25. The fat content was determined from a 1 g sample using the 2:1 (*v*/*v*) chloroform/methanol rapid solvent extraction method.

### 2.4. Analysis of Fatty Acids

The liquid in the *longissimus dorsi* was extracted following the method of Folch et al. [14]. Briefly, 5 g of crushed mutton was mixed with 5 mL of a 0.5 mol/L NaOH/methanol solution at 70 °C for approximately 5 min. Then, 5 mL of 14% boron trifluoride in methanol was added to the mixture at 70 °C for 5 min and mechanically shaken for 10 min at 70 °C. Subsequently, the mixture was combined with 5 mL of saturated NaHCO_3_ and 3 mL of isobutane at room temperature for about 30 min. The final supernatants were used to determine individual fatty acids. The separation was carried out using an Agilent7890A gas chromatograph (Agilent, Santa Clara, CA, USA) using an Rt-2560 fused-silica capillary column (100 m × 0.25 mm i.d. × 0.20 μm film thickness; Restek Corporation, Pennsylvania, PA, USA) and a flame ionization detector. The injector and detector temperatures were 250 °C and 280 °C, respectively. The column oven temperature was held at 120 °C for 5 min and programmed to increase to 250 °C at 4 °C/min and held for 28 min. Nitrogen was used as the carrier gas at a pressure of 30 psi.

### 2.5. Preparation of Longissimus Dorsi for LC-MS and Identification of Compounds

The preparation of the *longissimus dorsi* was extracted by LC-MS according to the method of Want et al. [15]. The *longissimus dorsi* was grounded in liquid nitrogen and placed in an EP tube. Then, 500 μL of 80% methanol was added to each sample aliquot and incubated for 5 min on ice. Then, the samples were centrifuged at 15,000× *g* and 4 °C for 5 min, and the supernatants were diluted with spectrometer-grade water until the methanol content was 53%. Subsequently, samples were analyzed by a gas chromatograph system (Vanquish UHPLC, Thermo Fisher, Gemering, Germany) coupled with a Q Exactive™ HF mass spectrometer (Q Exactive™ HF, Thermo Fisher, Bremen, Germany; LC-MS). The system utilized a Hypesil Gold column (100 × 2.1 mm inner diameter, 1.9 μm film thickness; Thermo Fisher, Waltham, MA, USA) using a 17 min linear gradient at a flow rate of 0.2 mL/min. The eluents for the positive-polarity mode were eluent A (0.1% FA in water) and eluent B (methanol). The eluents for the negative-polarity mode were eluent A (5 mM ammonium acetate, pH 9.0) and eluent B (methanol). The solvent gradient was set as follows: 2% B, 1.5 min; 2–100% B, 3 min; 100% B, 10 min; 100–2% B, 10.1 min; and 2% B, 12 min. The Q Exactive^TM^ HF mass spectrometer was operated in positive/negative-polarity mode with a spray voltage of 3.5 kV, a capillary temperature of 320 °C, a sheath gas flow rate of 35 psi, an aux gas flow rate of 10 L/min, an S-lens RF level of 60, and an aux gas heater temperature of 350 °C.

For metabolomic data, the metaX software was used to perform a principal component analysis (PCA) [16]. Differentially expressed metabolites were chosen for the following analysis with Variable Importance in Projection (VIP) > 1. The T-test was used to identify significantly different metabolites (*p* > 0.05). For clustering heat maps, the data were normalized using z-scores of the intensity areas of differential metabolites and were plotted by the Pheatmap package in R language version 3.4.3. The functions of these metabolites and metabolic pathways were studied using the KEGG database. The metabolic pathway enrichment of differential metabolites was analyzed: when the ratio satisfied x/n > y/N, the metabolic pathway was considered to be enriched; when the *p*-value of the metabolic pathway was <0.05, the metabolic pathway was considered statistically significantly enriched.

### 2.6. Statistical Analysis

The muscle fiber characteristics, chemical components, and fatty acids were analyzed by using SAS version 9.4 (SAS Institute Inc., Cary, NC, USA). The PROC TTEST procedure was used to calculate the differences between Black Tibetan and Chaka sheep. In all analyses, statistical significance was assumed at *p* < 0.05, and trends were declared at 0.05 < *p* < 0.10. Correlations between differential metabolites were analyzed using the Pearson method in R language version 3.4.3. Statistically significant correlations between differential metabolites were calculated by cor.mtest in R language version 3.4.3. A *p*-value < 0.05 was considered statistically significant, and correlation plots were plotted by the corrplot package in R language version 3.4.3.

## 3. Results

### 3.1. Muscle Fiber Characteristics

The muscle fiber characteristics of Black Tibetan and Chaka sheep are shown in Table 1 and Figure 1. The diameter (42.23 vs. 51.46 μm), perimeter (131.78 vs. 166.14 μm), and area of muscle fibers (1328.74 vs. 1998.64 μm^2^) were smaller in Chaka sheep than in Black Tibetan sheep.

### 3.2. Chemical Components of Mutton

The chemical components of the *longissimus dorsi* are illustrated in Table 2. The contents of DM, EE, and CP in the *longissimus dorsi* showed no differences between Black Tibetan and Chaka sheep (*p* > 0.05). The ash content in the *longissimus dorsi* was lower in Chaka sheep than in Black Tibetan sheep (*p* = 0.010).

### 3.3. Fatty Acids

The differences in fatty acids in the *longissimus dorsi* between Black Tibetan and Chaka sheep are presented in Table 3. The proportions of C10:0, C15:0, and tC18:1 was higher in Chaka sheep than in Black Tibetan sheep (*p* < 0.05). However, all other individual fatty acids were similar among treatments (*p* > 0.05). SFAs, UFAs, MUFAs, PUFAs, and the ratios of n-6 PUFAs to n-3 PUFAs and PUFAs to SFAs in the *longissimus dorsi* did not differ between Black Tibetan and Chaka sheep (*p* > 0.05).

### 3.4. PCA Analysis

The PCA score plot shows that the two groups were separated into independent clusters (Figure 2). The two principal components accounted for 26.51% (PC1) and 22.23% (PC2) of the explained variance.

### 3.5. Metabolic Biomarkers

A total of 65 biomarkers were identified between the two breeds of sheep (Figure 3). The metabolites mainly include 30 organic acids, 9 amino acids, 5 peptides, 4 amides, 3 adenosines, 2 amines, and other compounds. Among these metabolites, 40 metabolic biomarkers were upregulated in the CK group compared to the BT group. However, 25 metabolic biomarkers were downregulated in the CK group compared with the BT group.

### 3.6. Correlation Analysis

In addition, there were complex co-occurrences and co-exclusions in these metabolic biomarkers (Figure 4). There were the following positive correlations: (11E,15Z)-9,10,13-trihydroxyoctadeca-11,15-dienoic acid and ACar 17:0 was positively correlated with adenylosuccinic acid, adenosine 5’-monophosphate, XLR11 N-(2-fluoropentyl) isomer, adenosine, ACar 13:0, ACar 18:0, 10-Hydroxydecanoic acid, and stearic acid. Moreover, there were the following negative correlations: 2-(2-amino-3-methylbutanamido)-3-phenylpropanoic acid was negatively correlated with adenosine 5’-monophosphate, adenosine, XLR11 N-(2-fluoropentyl) isomer, ACar 13:0, ACar 18:0, 10-Hydroxydecanoic acid, stearic acid, adenylosuccinic acid, oleamide, XLR11 N-(2-fluoropentyl) isomer, 10-Hydroxydecanoic acid, and D-Ala-D-Ala.

### 3.7. KEGG Pathways

Based on KEGG pathway analysis, the metabolic biomarkers are involved in 20 functional pathways, such as fatty acid biosynthesis and purine metabolism (Figure 5). Among these pathways, eight pathways, namely, fatty acid biosynthesis, purine metabolism, the biosynthesis of unsaturated fatty acids, renin secretion, the regulation of lipolysis in adipocytes, neuroactive ligand–receptor interaction, the cGMP-PKG signaling pathway, and the cAMP signaling pathway, were identified as significantly different pathways.

## 4. Discussion

Muscle fibers, as the basic muscle units, directly affect meat quality, especially its tenderness, flavor, and juiciness [17]. In the present study, it was found that the diameter, perimeter, and area of muscle fibers were smaller in Chaka sheep than in Black Tibetan sheep. Studies have identified that breeds within a species are the most crucial factor influencing the compositions of muscle fiber characteristics, including the type, size, and total number of muscle fibers [18]. However, Bünger et al. reported that the average muscle fiber size was similar between the breeds [19]. The differences in the muscle fiber size in this study were likely due to a shift in the frequency of different fiber types, namely, an increase in fast fibers, a decrease in slow fibers, and a strong decrease in oxidative fibers [19].

The breed has a strong effect on the chemical composition of mutton, such as moisture, protein, fat, and ash [20]. Earlier studies found that the meat from goat carcasses contained higher moisture content and lower EE than sheep, whereas protein and ash contents were similar between the two species [21]. Nevertheless, our findings showed that the amount of ash in mutton was lower in Chaka sheep than in Black Tibetan sheep. The results indicated that mutton from Black Tibetan sheep had a higher mineral content. However, there were no differences in the DM, EE, and CP contents of mutton between Black Tibetan and Chaka sheep. These results also agree with the present findings. Lee et al. reported that no differences were found in moisture, protein, and fat percentages between the *longissimus dorsi* muscles of goats and lambs [22]. These results were attributed to the fact that sheep consumed food from the same pasture on the Qinghai-Tibet Plateau. According to different studies, there are many factors that are often difficult to control between studies, including differences in the maturity, breed, production system, and feeding management of animals used in these studies [21].

Nutritionists recommend increasing the intake of n-3 PUFAs in the human diet, and mutton is one of the major sources of these fatty acids. Earlier studies reported that pasture-fed sheep had higher contents of PUFAs than those of sheep raised indoors and were more conducive to consumer health [23]. In the present study, there were no differences in MUFA, PUFA, and SFA contents in the *longissimus dorsi* between Black Tibetan and Chaka sheep. In contrast, Mezöszentgyörgyi et al. found that Booroola Merino sheep contained a higher proportion of total SFAs and a lower proportion of TUFAs in their adipose tissue than Suffolk and Pannon sheep [24]. Van Harten et al. reported that Damara sheep with a restricted feed diet had higher PUFAs and lower SFAs and MUFAs compared with Merino and Dorper sheep [25]. On the other hand, Lopes et al. demonstrated that different goat genotypes displayed minor differences in the quality of their meat and fatty acid profiles [26]. Similarly, Smith et al. also reported that there was a small difference in the fatty acid composition of beef from Bos indicus and Bos taurus cattle [27]. According to other studies, one explanation for this difference is likely to be the different regimes, such as traditional grazing versus stall feeding, and different dietary compositions. These results suggest that dietary factors are considerably more prominent than the genotype in determining the fatty acid composition, particularly UFAs.

Studies have shown that the proportions of C10:0, C15:0, and tC18:1 are higher in Chaka sheep than in Black Tibetan sheep. The results are consistent with Gonzales-Barron et al., who found that the proportion of C15:0 was higher in Sambucana lamb meat (0.66%) compared with the other two breeds [28]. In contrast, Raes et al. reported that the tC18:1 fatty isomer was enriched in digestive contents [29]. Compared to rumen contents, subcutaneous and intramuscular fat contained a lower proportion of the tC18:1 isomer [29]. The difference in tC18:1 in Chaka sheep was likely due to the increase in Δ^9^-desaturase.

PCA was performed using metabolic data to determine whether the meat samples could be differentiated. In the current study, based on PCA analysis, it was found that the metabolomic profiles were significantly different between Black Tibetan and Chaka sheep, indicating that the metabolites were significantly different between the two breeds of sheep. A total of 65 metabolic biomarkers were found between Black Tibetan and Chaka sheep. The main metabolites include 30 organic acids, 9 amino acids, 5 peptides, 4 amides, 3 adenosines, 2 amines, and other compounds. Among these metabolites, 40 metabolic biomarkers were upregulated in Chaka sheep compared to Black Tibetan sheep, while 25 metabolic biomarkers were downregulated.

Metabolic biomarkers can reveal the different metabolites between Black Tibetan and Chaka sheep. To further analyze the underlying metabolic pathways in Black Tibetan and Chaka sheep, KEGG was used to analyze the functions of differential metabolite-related pathways, and 65 differential metabolites were enriched in 20 metabolic pathways. Based on metabolomics, it was found that the main enriched pathways were fatty acid biosynthesis, purine metabolism, the biosynthesis of unsaturated fatty acids, renin secretion, the regulation of lipolysis in adipocytes, neuroactive ligand–receptor interaction, the cGMP-PKG signaling pathway, and the cAMP signaling pathway. In terms of further pathway enrichment, fatty acid biosynthesis had the largest impact among the different pathways. Lauric acid, decanoic acid, palmitic acid, and stearic acid have been established as key players in this pathway and were increased in Chaka sheep. Lauric acid and palmitic acid are considered to be detrimental to health and to be lower in lambs finished in pastures [30]. These results indicate that there is a difference in the synthesis of SFAs between the two breeds of sheep.

A study demonstrated that the flavor of Tibetan sheep could be increased by regulating purine metabolism [31]. Purine metabolites provide a cell with the necessary energy and cofactors to promote cell survival and proliferation [32]. In the present study, adenosine, adenosine 5’-monophosphate, and adenylosuccinic acid were enriched in the pathway of purine metabolism, and these metabolites were upregulated in Chaka sheep compared to Black Tibetan sheep. The results showed that (11E,15Z)-9,10,13-trihydroxyoctadeca-11,15-dienoic acid and ACar 17:0 had positive correlations with adenylosuccinic acid, adenosine 5’-monophosphate, XLR11 N-(2-fluoropentyl) isomer, adenosine, ACar 13:0, ACar 18:0, 10-Hydroxydecanoic acid, and stearic acid. This is because most of the organic acids in sheep meat are degradation products of fat. 2-(2-Amino-3-methylbutanamido)-3-phenylpropanoic acid displayed negative correlations with these organic acids. Evidence has shown that some of them are precursors of the Maillard reaction, and some are produced by the Strecker amino acid degradation reaction [33]. Adenylosuccinic acid is cleaved by adenylosuccinic acid lyase to form adenosine 5’-monophosphate (AMP) [34]. As a consequence, this result suggests that the cells in Chaka sheep’s *longissimus dorsi* have improved energy utilization and proliferation that regulate the flavor of mutton. It was found adenosine and adenosine 5’-monophosphate were enriched in the pathway of lipolysis regulation in adipocytes. It is possible that this metabolically necessary energy was directed to lipolysis in adipocytes. Adenosine combines with adenosine receptor A1, which plays a vital role in lipolysis in the adipocyte pathway [35]. Similarly, taurine was bound to glycine receptor alpha-1 to regulate membrane trafficking. Studies have identified that taurine can affect systemic energy metabolism, specifically lipid metabolism, since an increase in lipid oxidation may promote carbohydrate savings [36]. Thus, it is speculated that the lipid metabolism in Chaka sheep is greater than that in Black Tibetan sheep.

Studies have shown that adenosine 5’-monophosphate is enriched in the cGMP-PKG signaling and cAMP signaling pathways, and it was upregulated in Chaka sheep. These findings mean that the energy efficiency of Chaka sheep was higher than that of Black Tibetan sheep. Among these pathways, it was found that epinephrine was downregulated in Chaka sheep compared to Black Tibetan sheep. Epinephrine stimulates lipolysis by binding to both α- and β-adrenergic receptors on the adipocyte cell surface, thereby increasing intracellular concentrations of cAMP and activating cAMP-sensitive (hormone-sensitive) lipase, which cleaves triglycerides into free fatty acids and glycerol [37]. The results suggest that Chaka sheep have the potential to prevent lipolysis decomposition.

In the present study, there was a difference in the regulation of lipolysis in adipocytes between Black Tibetan and Chaka sheep. Among these metabolites, the large variation in fatty acids in the present study indicates that fatty acids are highly sensitive to the breed. We found that the metabolites ACar 10:1, ACar 13:0, ACar 15:0, ACar 17:0, ACar 18:0, ACar 18:1, ACar 19:0, ACar 19:1, ACar 20:0, ACar 20:2, and ACar 20:3 was increased in Chaka sheep. This observation is in agreement with the findings that lipid metabolism in Chaka sheep was greater than in Black Tibetan sheep. The above findings indicate an increase in fatty acid biosynthesis in Chaka sheep compared with Black Tibetan sheep.

Bioactive peptides can be used as basic compounds in functional foods [38]. Peptides with 2-6 amino acids are absorbed more readily compared to proteins and free amino acids [39]. Among these metabolites, it was found that Gly-Tyr, Asp-Phe, and Gly-Phe were downregulated in Chaka sheep compared to Black Tibetan sheep. Additionally, Sarower et al. reported that glycine and phenylalanine are important contributors to the taste of raw meat [40]. The peptides Gly-Tyr and Gly-Phe have a bitter taste, while Asp-Phe has a sour taste [40]. Based on these findings, our results indicate that these peptides play a key role in the flavor of mutton.

## 5. Conclusions

In summary, we compared the muscle fiber characteristics, fatty acid profiles, and metabolites in mutton between Black Tibetan and Chaka sheep. Chaka sheep exhibited higher proportions of C10:0, C15:0, and tC18:1 in the *longissimus dorsi* and a lower content of ash. A total of 65 metabolic biomarkers were identified between the two breeds of sheep. Among these metabolites, 40 metabolic biomarkers were upregulated in Chaka sheep, and 25 metabolites were downregulated. The main metabolites include 30 organic acids, 9 amino acids, 5 peptides, 4 amides, 3 adenosines, 2 amines, and other compounds. Based on KEGG analysis, eight pathways, namely, fatty acid biosynthesis, purine metabolism, the biosynthesis of unsaturated fatty acids, renin secretion, the regulation of lipolysis in adipocytes, neuroactive ligand–receptor interaction, the cGMP-PKG signaling pathway, and the cAMP signaling pathway, were identified as significantly different pathways. According to the results on fatty acids and metabolites, upregulated organic acid and fatty acid biosynthesis increased the meat quality of Chaka sheep.

## Figures and Tables

**Figure 1 animals-12-02745-f001:**
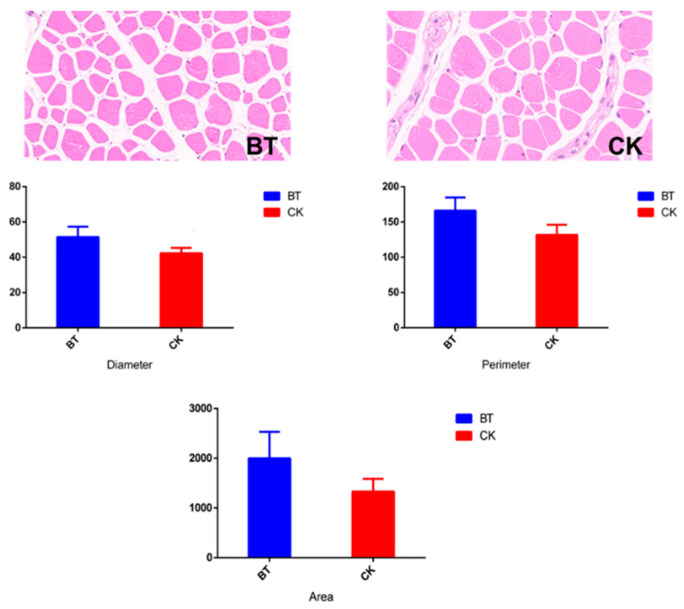
The muscle fiber characteristics in *l**ongissimus dorsi* muscles. BT, Black Tibetan sheep group; CK, Chaka sheep group.

**Figure 2 animals-12-02745-f002:**
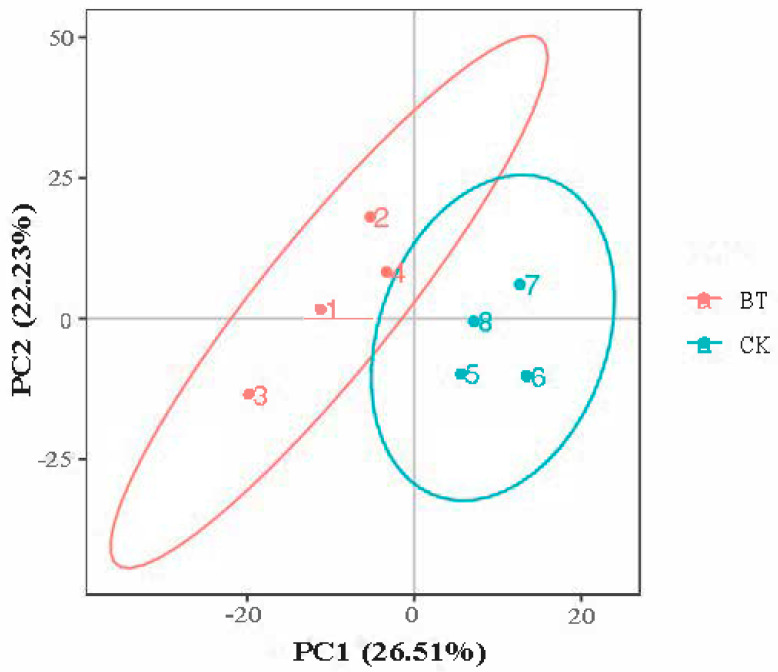
PCA shows differences in metabolic profiles between Black Tibetan and Chaka sheep. BT, Black Tibetan sheep group; CK, Chaka sheep group.

**Figure 3 animals-12-02745-f003:**
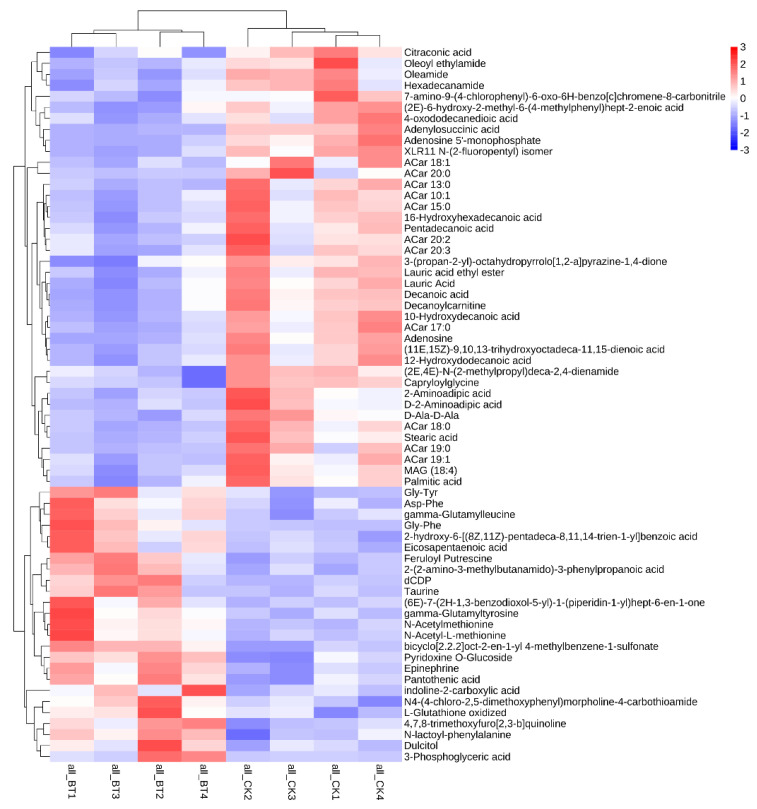
Differences in metabolic profiles between Black Tibetan and Chaka sheep.

**Figure 4 animals-12-02745-f004:**
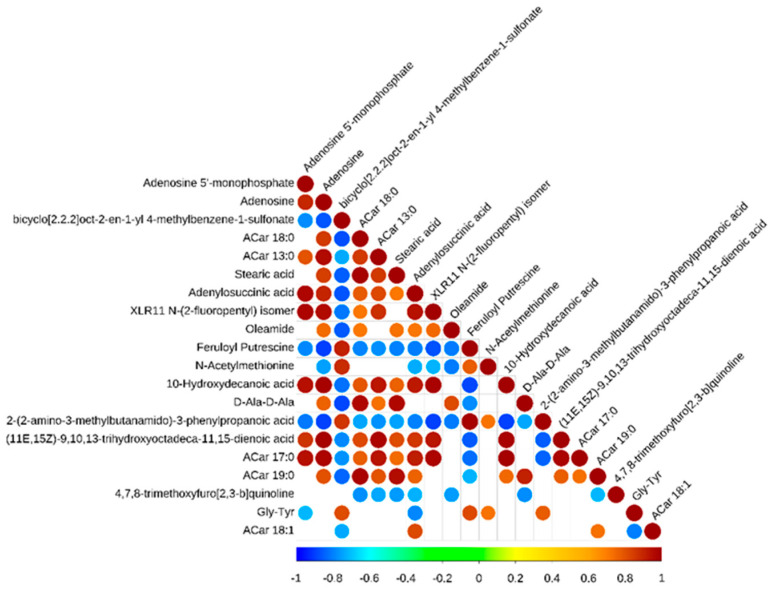
Correlation heat map of differences in metabolic biomarkers between Black Tibetan and Chaka sheep.

**Figure 5 animals-12-02745-f005:**
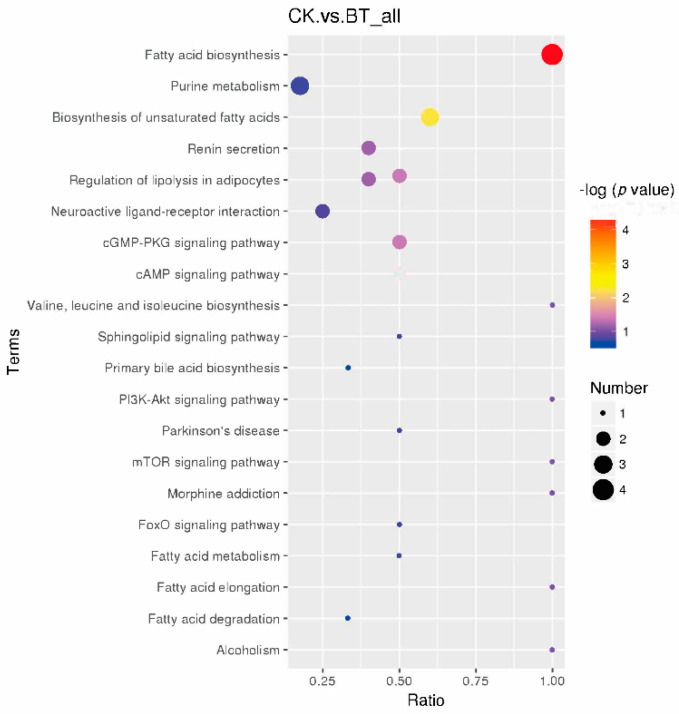
KEGG pathways of different metabolites in the *longissimus dorsi* of Black Tibetan and Chaka sheep.

**Table 1 animals-12-02745-t001:** The muscle fiber characteristics of Black Tibetan and Chaka sheep.

	BT ^1^	CK ^2^	SEM ^3^	*p*-Value
Diameter, μm	51.46 ± 5.867	42.23 ± 3.102	3.319	0.032
Perimeter, μm	166.14 ± 18.604	131.78 ± 14.362	11.751	0.026
Area, μm^2^	1998.64 ± 531.776	1328.74 ± 257.096	295.332	0.064

^1^ BT, Black Tibetan sheep group; ^2^ CK, Chaka sheep group; ^3^ SEM, standard error of the mean.

**Table 2 animals-12-02745-t002:** The proportions of basic chemical components in mutton (DM basic, %).

	BT ^1^	CK ^2^	SEM ^3^	*p*-Value
DM	26.57 ± 0.473	27.38 ± 1.589	0.958	0.452
Ash	2.27 ± 0.235	1.71 ± 0.192	0.151	0.010
EE	3.51 ± 0.184	5.07 ± 2.132	1.070	0.239
CP	19.87 ± 0.598	19.13 ± 0.668	0.448	0.145

^1^ BT, Black Tibetan sheep group; ^2^ CK, Chaka sheep group; ^3^ SEM, standard error of the mean.

**Table 3 animals-12-02745-t003:** The contents of individual fatty acids and calculated fatty acids.

(%)	BT ^1^	CK ^2^	SEM ^3^	*p*-Value
C10:0	0.02 ± 0.004	0.05 ± 0.014	0.007	0.036
C14:0	0.43 ± 0.086	0.48 ± 0.213	0.115	0.686
C15:0	0.24 ± 0.021	0.28 ± 0.011	0.012	0.030
C15:1	6.63 ± 0.671	6.21 ± 1.112	0.650	0.549
C16:0	13.95 ± 0.619	15.27 ± 1.571	0.845	0.168
C16:1	2.11 ± 0.138	2.09 ± 0.135	0.097	0.804
C17:0	1.05 ± 0.110	1.05 ± 0.050	0.060	0.937
C17:1	6.50 ± 0.314	5.71 ± 0.884	0.469	0.143
C18:0	10.48 ± 0.512	11.90 ± 2.581	1.315	0.353
tC18:1	0.76 ± 0.098	1.12 ± 0.234	0.778	0.030
C18:1	25.98 ± 2.447	25.68 ± 2.878	1.315	0.614
tC18:2	0.33 ± 0.046	0.42 ± 0.149	0.078	0.290
C18:2	10.80 ± 0.632	10.10 ± 2.552	1.315	0.614
C18:3	2.02 ± 0.219	1.69 ± 0.393	0.225	0.192
C20:1	0.16 ± 0.053	0.19 ± 0.138	0.074	0.751
C20:3	5.28 ± 0.674	5.26 ± 0.994	0.600	0.975
C20:5	2.01 ± 0.256	1.59 ± 0.409	0.242	0.141
C22:6	0.72 ± 0.120	0.69 ± 0.233	0.131	0.800
SFAs ^4^	26.17 ± 0.732	29.03 ± 4.259	2.160	0.235
UFAs ^5^	63.29 ± 0.787	60.74 ± 3.418	1.753	0.196
MUFAs ^6^	42.14 ± 2.001	40.99 ± 1.131	1.149	0.358
PUFAs ^7^	20.83 ± 1.358	19.33 ± 4.444	2.323	0.544
n-6 PUFAs	11.52 ± 0.667	10.79 ± 2.748	1.414	0.622
n-3 PUFAs	9.30 ± 0.916	8.54 ± 1.726	0.976	0.466
n-6/n-3	1.25 ± 0.118	1.25 ± 0.115	0.082	0.951

^1^ BT, Black Tibetan sheep group; ^2^ CK, Chaka sheep group; ^3^ SEM, standard error of the mean; ^4^ SFAs, saturated fatty acids; ^5^ UFAs, unsaturated fatty acids; ^6^ MUFAs, monounsaturated fatty acids; ^7^ PUFAs, polyunsaturated fatty acids.

## Data Availability

Not applicable.

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
