# Peer review of "Comparative Analysis of the Composition of Fatty Acids and Metabolites between Black Tibetan and Chaka Sheep on the Qinghai—Tibet Plateau"

_animals, 2022, doi:10.3390/ani12202745_

Round 1

Reviewer 1 Report

The article reported on the composition difference of fatty acids and metabolites between two sheep breeds in Qinghai-Tibetan Plateau. The study was well designed, comprehensive and provided useful data for both animal breeders and researchers in the field of sheep industry. The results are presented in a clear and concise manner and the authors do a good job of putting their results in the context of previous work. I found the functional enrichment analysis to be of particular interest and think these results will be well used for livestock industry. I only have a few minor points to improve the manuscript.

There are a few minor details that needs editing, but overall I think the paper can be accepted for publication after revision.

1. There may be some mistakes in the description of Chaka sheep and Black-Tibetan sheep in summary part. As we known, Chaka sheep is a semi-wool cultivated breed and Black-Tibetan, which is also named as Guide Black Fur Sheep, is a native breed. Please check that.

2. Please rewrite the materials and methods part. The metabolites can be affected by lots of factors, such as age. Please note the age of the selected animals. HE staining was a conventional method that you can describe briefly or ignore it. But I suggest you provide more detailed information about LC-MS such as flow rate and solvent of the gradient. There may be a mistake about the chromatograph system in Line 135. Please provide the version of R used in the analysis.

3. The number 2 in μm2 should be superscribed in Line 164.

4. I strongly suggested the authors described the metabolites differed in BT and CK groups into different clusters, such as organic acids, lipids, amino acids etc, but not listed it in the manuscript simply.

5. It will be more interesting to find the fatty acids related metabolites with correlation analysis. 

Reviewer 2 Report

This was studied to investigate the fatty acids and metabolites in longissi mus dorsi muscles between Chaka sheep and Black-Tibetan sheep grazing in a highly saline environment, there are some studies have reported the similar papers, please explain the innovativeness. The other, the language needs to be improved.

1.     Abstract is also not reflect the content of the article and needs to be rewritten.

2.     The aim of this paper is not clear.

3.     Experimental Design:  The sentences need improve.

4.     More information of sheeps are not also provided, such as protein, water and so on.

5.     Statistical Analysis need to be rewritten.

6.     Disscussion need improve.

7.     Conclusions need rewrite.

Reviewer 3 Report

General question. Why fatty acids composition has been investigated in longissimus dorsi? Fat content in this part may variate from 2 to 22% depending on cutting. Wouldn’t it be more useful and more indicative to do it in fat tail of sheep?

There is information in chapter 2.2 about analysis of nutrition composition, but the results of this analysis have not been given in article. Therefore authors should add that results or delete the information about the methods of nutrition composition investigation.

Lines 99-100. According to the information in these lines, meat was used right after slaughtering and freezing afterwards for this research. No information about meat aging was given. Autolysis process plays significant role in formation of meat characteristics. Autolysis also affects fatty acids compounds because of fat oxidation during it. So authors should clearise this moment about meat aging, whether it was or not.

Lines 27, 31, 261, 288 etc. “Was decreased, was increased” better be changed to “was lower, was higher”, because it’s more applicable for comparison.

Lines 260, 305, 331 etc. Articles should be written using impersonal sentences. So parts such as “we found…, in our study…” better be changed to “studies have shown…, it was found…”

Round 2

Reviewer 2 Report

The language also needs to be improved. Such as:

L24-25 The experiment lasted for 20 months All sheep grazed in a highly saline environment in a whole experiment period and had free access to water. ?

L166 ()?
